# The Relationship between Patients’ Demands and Workplace Violence among Healthcare Workers: A Multilevel Look Focusing on the Moderating Role of Psychosocial Working Conditions

**DOI:** 10.3390/ijerph21020178

**Published:** 2024-02-04

**Authors:** Cristian Balducci, Chiara Rafanelli, Luca Menghini, Chiara Consiglio

**Affiliations:** 1Department of Quality of Life Sciences, University of Bologna, 47921 Rimini, Italy; 2Department of Psychology “Renzo Canestrari”, University of Bologna, 40127 Bologna, Italy; chiara.rafanelli@unibo.it; 3Department of General Psychology, University of Padova, 35131 Padova, Italy; luca.menghini@unipd.it; 4Department of Psychology, Sapienza University of Rome, 00185 Rome, Italy

**Keywords:** workplace violence, patients’ demands, psychosocial working conditions, workload, job control, supervisor support

## Abstract

Workplace violence against healthcare workers is a widespread phenomenon with very severe consequences for the individuals affected and their organizations. The role played by psychosocial working conditions in healthcare workers’ experiences of violence from patients and their family members has received relatively scant attention. In the present study, we investigated the idea that psychosocial working conditions (workload, job control, supervisor support, and team integration), by affecting the well-being and job performance of healthcare workers, play a critical role in the relationship between patients’ demands and the escalation of workplace violence. Specifically, we tested the hypothesis that psychosocial working conditions moderate the relationship between patients’ demands and workplace violence. Participants were 681 healthcare workers distributed in 55 work groups of three public healthcare facilities in Italy. Multilevel analysis showed significant interactions between patients’ demands and each of the investigated psychosocial factors on workplace violence, which in all the cases were in the expected direction. The results suggest that improving the quality of the psychosocial work environment in which healthcare workers operate may be a critical aspect in the prevention of workplace violence.

## 1. Introduction

Workplace violence is a severe and widespread phenomenon in today’s workplaces [1]. It regards incidents where staff are abused, threatened, or assaulted in circumstances related to their work [2]. The consequences for the individuals exposed to workplace violence may be dramatic, including an increased risk for suicide [3,4,5]. At the organizational level, the impact of workplace violence is equally catastrophic, resulting in prolonged sickness absence, intention to leave the organization, and actual turnover [6,7,8]. There has been a substantial increase in attention to the phenomenon over the past 20 years or so, leading to initiatives such as the recent International Labor Organization (ILO) ‘Violence and Harassment Convention 190’ in 2019, which urges member states to enact a comprehensive strategy aimed at preventing such negative behavior within the workplace.

Among different work sectors, the healthcare sector reports the highest prevalence of workplace violence. For example, at the European Union level, 20 percent of healthcare workers reported having been the subject of verbal abuse in the past month, and seven percent indicated that they were victims of physical violence [9], suggesting that workplace violence is a common occurrence for this occupational group, almost a ‘normal part’ of the job [10]. Additionally, the most frequent form of violence in the sector is third-party violence, that is, violence perpetrated in most cases by patients and their family members. This accounts for up to 93 percent of all assaults against workers [11].

Given the above state of affairs, it is imperative to better understand the antecedents of workplace violence so that specific preventive interventions can be implemented. Available evidence has indicated that certain patient characteristics, such as experiencing pain, difficult prognoses, and mental disorders, may act as significant predictors of workplace violence towards hospital staff [12,13]. Among work environmental and organizational conditions, prior research has mainly focused on physical workplace characteristics such as the inadequate physical layout of buildings, the lack of alarm systems (e.g., metal detectors), and other factors including long waiting times and understaffing [12,14]. Although psychosocial factors have also been considered [15], attention to them has been relatively scant, with some authoritative guidelines for preventing workplace violence in the healthcare sector totally ignoring them [16].

Psychosocial factors have traditionally been considered for their impact on workers’ job stress and mental health [17], while less attention has been paid to the potential of certain psychosocial working conditions to be directly involved in the dynamic of third-party workplace violence, which may be—at least in part—a side effect of a poor psychosocial work environment [18,19]. Therefore, the present study was designed with the main aim of assessing the role of psychosocial working conditions in reports of workplace violence from patients and their family members, building on the idea that violence is frequently an escalated outcome of demands from patients and their family members, which at times may be excessive and unrealistic [20,21]. Specifically, a violent episode may develop from the inadequate management of patients’ demands, which may be significantly impacted by poor psychosocial working conditions fueling job stress and reducing the effectiveness of healthcare workers’ job performance. For example, patients may expect nurses to satisfy their wishes (e.g., reduce their waiting time) without considering their work-related constraints. Denial of such requests may lead to frustration, which is a powerful mechanism for the enactment of aggression [22]. Under such circumstances, managing interactions with patients and their relatives requires calmness and balance and, more in general, a psychological state that has been described as “detached concern”—being completely “there” for the patient but at the same time being sufficiently detached from the patient’s emotional state [23]. However, if healthcare workers are exposed to psychosocial risk factors related to a poorly organized work environment, they will experience high levels of stress, which is incompatible with the state of calmness and balance needed to manage patients’ demands, especially when they are excessive.

High levels of stress go hand in hand with the experience of negative affective states such as tension and anxiety, [24]. Similarly, chronic stress may, in the longer run, evolve into a cynical attitude towards the patient and impaired emotional control as part of the burnout syndrome [25,26,27] that disconnects the individual from others, leading to overreaction and patient blaming [28]. Such psychological states and conditions may fuel patients’ irritability, increasing the likelihood of violent incidents.

### Study Hypotheses

To test the above ideas, we adopted a multilevel perspective in the present study, considering work environmental psychosocial risk factors acting at both individual and group levels. At the individual level, we focused on high workload (quantitative demands) and job control. These two factors have received extensive attention as part of well-established job stress models [24,29]. Workload, concerning the amount of work that has to be carried out within the available time, is the stressor ‘par excellence’, and a high workload is commonly reported among healthcare workers [30]. Therefore, when healthcare workers face high workload levels, they also experience common manifestations of job stress (tension and hostility) that may amplify the link between excessive patients’ demands and workplace violence.

**Hypothesis** **1 ****(H1).**
*The relationship between excessive patients’ demands and workplace violence will be stronger when workload is high.*


On the contrary, the availability of more control, such as having a voice in the organization of one’s work and being able to influence the work pace and shifts, may act as an important job resource by fulfilling the basic human need for autonomy [31], which protects against the experience of job stress. Therefore, consistent with what has been found in previous research on the protective role of control [32], high levels of control should weaken the relationship between patients’ demands and workplace violence.

**Hypothesis** **2**** (H2).**
*The relationship between excessive patients’ demands and workplace violence will be weaker when job control is high.*


At the group level, we considered two factors that comparably apply to all workers exposed to the same work environment, namely supervisor support and team integration. Supervisor support is a powerful resource in reducing stress [33,34]. For example, a supervisor that, during a difficult exchange between a worker and an angry patient, steps in and is available to mediate the conflict, or that, more generally, is socially accessible, contributes to building a better psychosocial safety climate at work. This can reduce the impact of stress-inducing experiences in the group, facilitating workers to maintain the necessary psychological flexibility and balance to deal with excessive patients’ demands [35,36]. Consequently, we hypothesized that supervisor support at the group level will reduce the within-group relationship between patients’ demands and workplace violence, with the relationship being weaker in those groups with higher levels of supervisor support.

**Hypothesis** **3**** (H3).**
*The relationship between excessive patients’ demands and workplace violence will be weaker when group-level supervisor support is high.*


A similar resource at the group level is what we define as ‘team integration’, expressing the degree to which constructive collaboration among different professionals (i.e., doctors, nurses, other personnel) is considered important and the extent to which the different professionals actively engage in complementing each other to reach common goals [37]. Such aspects, which characterize effective cross-functional teamwork, tend to generate a protective psychological atmosphere at work and a supportive environment that may act as a crucial organizational social resource against the experience of job stress [35]. Therefore, we expected that team integration moderates the relationship between patients’ demands and workplace violence, with a weaker relationship in the groups with a higher level of integration.

**Hypothesis** **4**** (H4).**
*The relationship between excessive patients’ demands and workplace violence will be weaker when group-level team integration is high.*


## 2. Materials and Methods

### 2.1. Participants and Procedure

Data collection took part in the context of an Italian national project funded by the Ministry of Health. The aim of the project was to develop updated guidelines and context-specific measuring instruments for the assessment of work-related stressors and their consequences according to the Italian Health and Safety Law (Legislative Decree n. 81/2008). One of the sectors considered in the study was the healthcare sector. The project included a data collection that took place in three healthcare facilities located in different parts of Italy, all of which agreed to participate in the project. A self-reported questionnaire that included a variety of items and scales thought to capture both general (e.g., workload) and context-specific (e.g., workplace violence) stressors was administered to healthcare professionals (physicians, nurses, and other hospital staff). Additionally, the facilities were also asked to collect some ‘objective’ data, including ‘official’ episodes of workplace violence registered in the organizational records. Since the data were subsequently used by the facilities for the mandatory work-related stress risk assessment, no ethical approval was necessary. The study was conducted in line with the Helsinki Declaration, as well as the Italian data protection regulation (Legislative Decree n. 196/2003). The data considered here were derived from the responses to the self-report questionnaire, which was filled in anonymously by a total of 807 employees. Response rate was 79% (ranging from 22 to 100%) in the included departments/wards. The analyses reported below are based on the 661 participants with complete data on all the study variables, distributed across 56 work groups. Participants were females in 72.5% of cases and had an age distribution as follows: 8.3% up to 30 years, 66.9% between 31 and 50 years, and 24.8% more than 50 years. Regarding job role, 49.6% were nurses, 24.7% were doctors (this group also included a few biologists, chemists, and physicists), 14.7% were assistance operators, and the remaining 11.0% included obstetricians, technicians, and administrative staff. Almost all participants (98.9%) were of Italian nationality and had a permanent job contract (95.6%). Organizational tenure was, on average, 14 years (SD = 9.8 years). Participants worked in several different units within their organization, including intensive care, the emergency department, and general medicine.

### 2.2. Measures

Workplace violence from patients and their family members was measured by adapting three items from the ‘physically intimidating behaviour’ dimension of the Negative Acts Questionnaire-Revised (NAQ-R) [38] (e.g., “During work, I’m the subject of verbal violence and mistreatment from patients or their family members”). Responses were given on a five-point scale ranging from Never (1) to Always (5). Cronbach’s α was 0.86. Composite reliability (ω) [39] was good at both levels of analysis (within groups: ω_w_ = 0.83; between groups: ω_b_ = 0.95). Additionally, for a subset of groups (n. = 26), we had group-level data on the number of officially registered workplace violence episodes that happened over the previous year, derived from organizational records. These data were not used for hypothesis testing; however, we used them to check for convergence with the group average level of workplace violence, as reported by employees (see below).

Patients’ demands were measured using four items taken from the ‘Disproportionate Customer Expectations’ dimension of the Customer-related social stressors scale (CSS) [20] (e.g., “Some patients think they are more important than others”). Responses were given on a five-point scale ranging from ‘Never’ (1) to ‘Always’ (5). Cronbach’s α was 0.88. ω values were the following: ω_w_ = 0.83; ω_b_ = 0.98.

Workload, job control, and supervisor support were measured using the corresponding scales of the UK Health and Safety Executive (HSE) Stress Management Indicator Tool (SMIT) [40]. Specifically, workload was assessed by eight items (e.g., “I am pressured to work long hours”), job control by five items (e.g., “I have a choice in deciding how I do my work”), and supervisor support by five items (e.g., “I can talk to my line manager about something that has upset or annoyed me about work”), with the response options ranging from ‘Never’ (1) to ‘Always’ (5) for most of the items, and from ‘Totally disagree’ (1) to ‘Totally agree’ (5) for a subset of items. Cronbach’s α of the three scales was, respectively, 0.81, 0.75, and 0.82. ω values were the following: ω_w_ = 0.77, ω_b_ = 0.82, for workload; ω_w_ = 0.74, ω_b_ = 0.83, for job control; and ω_w_ = 0.82, ω_b_ = 0.90, for supervisor support.

Team integration was measured by four items [37] investigating the quality of cross-functional teamwork (“There is much collaboration between doctors, nurses and other professionals in the team”). Responses varied between ‘Never’ (1) to ‘Always’ (5). Cronbach’s α was 0.89, while ω values were the following: ω_w_ = 0.88; ω_b_ = 0.98.

### 2.3. Analysis

Participants were clustered within work groups; therefore, data were multilevel in nature and required the use of multilevel analysis [41], which we implemented by using SPSS-26 with the maximum likelihood estimator. For each main predictor (i.e., patients’ demands, workload, job control, team integration, and supervisor support) we derived the group mean centered version of the variable, which included only within-group variance. In addition, we also aggregated each predictor at the group level and obtained a second version of each variable including only group means (i.e., between-group variance only). In the main analyses, we used the group mean centered variable for patients’ demands, workload, and job control, while we used the aggregated variable (centered at the grand mean) for team integration and supervisor support, which we considered group characteristics. To test the study hypotheses, we fitted a series of multilevel models. We first estimated a null (intercept only) model. Then, we estimated a model (Model 1) including sociodemographic and occupational variables, patients’ demands (i.e., the main independent variable of the study), and the variables representing psychosocial working conditions acting as moderators at the individual and group levels. In Models 2 and 3, we added the same-level interactions between the main independent variable (patients’ demands) and workload and job control, respectively. In Model 4, we included two random components, that is, a random slope for patients’ demands and the covariance between intercept and slope. This model aimed to ensure that there was a significant amount of slope variance for patients’ demands. Finally, in Models 5 and 6, we added the cross-level interactions between patients’ demands and each of the two group-level predictors (i.e., supervisor support and team integration), respectively. Nested models were compared by conducting a −2Log likelihood difference (or deviance) test [41].

## 3. Results

Bivariate correlations between the main study variables are reported in Table 1.

Results showed that workplace violence from patients and their family members was positively and significantly associated, within groups, with patients’ demands (0.40, *p* < 0.001), indicating that healthcare workers reporting higher levels of excessive demands from patients compared to their group colleagues, also reported a higher level of exposure to workplace violence from patients and their family members. The between-group correlation between the two variables was even stronger (0.73, *p* < 0.001), suggesting that groups with a higher average level of excessive demands from patients were also characterized by a higher average level of workplace violence. Within groups, both workload and job control showed significant correlations in the expected direction with workplace violence. Similarly, at the group level, both supervisory support and team integration showed significant and negative correlations with workplace violence. The group average level of workplace violence was also positively and moderately associated with the indicator of the officially registered workplace violence episodes, that is ‘workplace violence (objective)’ (*r* = 0.42, *p* < 0.05), which demonstrates a certain degree of convergence between self-reported and ‘objective’ data.

The variance components in the Null model (see Table 2) led to an intraclass correlation coefficient (ICC) of 0.31, indicating that 31% of variance in workplace violence was related to differences between groups of workers (group level) and 69% to differences between workers within their group (individual level) Thus, there was a substantial amount of variance to be explained at both levels of analysis. Model 1 showed that patients’ demands were positively and significantly related with workplace violence (b = 0.35, *p* < 0.001) at the within-group level, over and above a series of control variables. We then examined the postulated interactions. Model 2 and Model 3 revealed, respectively, that the within-group relationship between patients’ demands and workplace violence was strengthened by workload (b = 0.22, *p* < 0.01), while it was attenuated by job control (b = −0.16, *p* < 0.05). The obtained interactions (i.e., same-level interactions) are graphically represented in Figure 1a,b—see below. In Model 4, we freed the slope of patients’ demands and allowed for the covariance between the random slope and the random intercept, estimating two additional parameters (i.e., random effects) compared to Model 1 (17 vs. 15—see Table 2). Model 4 showed a better fit than Model 1 (∆ −2 log likelihood (2) = 9.63, *p* < 0.001) and suggested that there was a sufficient amount of slope variance between the investigated groups. Model 5 showed that the within-group relationship between patients’ demands and workplace violence was significantly moderated by group-level supervisor support (b = −0.22, *p* < 0.01) (i.e., cross-level moderation), indicating that the relationship between patients’ demands and workplace violence was significantly weaker in groups with a higher average level of supervisor support than in groups with a lower level of support (see Figure 1c). Similarly, Model 6 indicated that the within-group relationship between patients’ demands and workplace violence was attenuated in those groups, characterized by a higher average level of team integration (b = −0.39, *p* < 0.001) (Figure 1d). Overall, we found clear support for the hypothesized moderations. 

## 4. Discussion

The results of our study support the idea that workplace violence from patients and their family members may be linked with an escalation initiated by excessive patients’ demands, the evolution of which is impacted by psychosocial working conditions at different levels (i.e., individual/job level and group/departmental level). Such psychosocial working conditions have the common denominator of being involved in the experience of job stress. Although a moderate level of stress may result in better job performance, sustained high levels of stress may significantly undermine healthcare workers’ well-being, eventually leading to a poorer management of the relationship with patients and their family members, which is a crucial aspect of task performance [42]. Indeed, job stress can arouse healthcare workers and alter their psychological experiences, fueling negative affective states such as tension, angry, and anxiety [24]. These states are incompatible with adopting a calm and balanced approach towards patients and their family members, particularly in the case of ‘difficult’ patients that find themselves in critical situations (i.e., experiencing physical pain, stress due to very long waiting times, etc.).

In line with the formulated hypothesis, we found that patients’ demands were more strongly related with the experience of workplace violence in healthcare workers who reported higher levels of workload. This suggests that the tension generated by frequently accomplishing a lot of tasks and engaging in fast-paced work activities may lead healthcare workers to show less concern for their patients, even showing signs of cynicism and withdrawal as a manifestation of chronic stress [30]. In other words, the moderating role highlighted for workload might indicate that higher workload can result in suboptimal performance in dealing with patients’ demands, accentuating the likelihood that excessive patient demands turn into episodes of mistreatment and violent behaviors from patients and/or their family members.

Similarly, but in the opposite direction, we found that a higher-than-average level of job control buffered the patient demands–workplace violence relationship within groups. In other words, the perceived availability of more autonomy in one’s job, which may manifest in different ways, such as having voice in the organization of one’s work and a certain degree of schedule flexibility and predictability, attenuated the link between excessive patients demands and workplace violence. It is well known that autonomy leads to greater self-determination, which is a basic human need that has been related to different facets of well-being, including lower stress levels and better job performance [31]. Indeed, experiencing higher levels of job control may lead to perceiving more responsibility for one’s work behavior, and this can generally fuel more positive job-related affective experiences as well as higher work engagement [31]. All this may bring benefits to the management of patients’ demands in different ways. For example, workers may dedicate more time, attention, and care to the specific needs of their patients, leading to higher patient satisfaction. Similarly, a better healthcare worker’s mood and a calm attitude as a consequence of more job control may be ‘contagious’ [43] and contribute to putting patients and their family members in better psychological conditions, defusing those critical internal states (e.g., hostile attributions, anger) that may act as immediate triggers of workplace aggression.

We also found that the potential of patients’ demands to result in violent behavior was attenuated by two distinct group-level resources, namely supervisor support and team integration. Results showed that in groups that benefited from more supervisor support, patients’ demands showed a weaker link with workplace violence, which is in line with the idea that supervisor support acts as a group-level buffer of workplace violence in response to patients demands. Supervisor support has been repeatedly shown to be of crucial importance for employee well-being, independently of their sector of employment [33,44]. For example, a study [33] showed that being unable to count on the support of a direct supervisor in case of problems at work, and even at home, involved a substantially increased risk (up to 3.8 times) of poor health and work-related outcomes including feeling overwhelmed at work and experiencing job dissatisfaction. Hämmig [33] concluded that, although multiple sources of social support are helpful, in particular, supervisor support seems to be an important resource for health and well-being at work and needs to be considered as a key factor in workplace health promotion. In line with our idea that workplace violence may result from an escalation of patients’ demands, the current findings suggest that such escalation is less likely to happen in groups/departments characterized by more supervisor support. All other factors being equal, these groups of workers might experience higher levels of well-being, which is of added value to managing difficult relationships with patients.

Another equally important resource at the group level, according to our results, is team integration, that is, the degree of cooperation and coordination between different professionals in healthcare teams/groups and the extent to which they complement and respect each other. Our results indicate that in groups characterized by better cross-functional teamwork, the relationship between patients’ demands and workplace violence is significantly attenuated. It is well-known that teamwork in the healthcare sector plays a crucial role in performance, including providing high-quality patient care [45], whereas deficiencies at this level may create tensions and stress among healthcare workers, with immediate repercussions on their relationships with patients and their family members. Additionally, low coordination, as indicated by poor teamwork, may be related with inconsistent behavior and communication towards patients, which may be detrimental for good exchanges with them. Conversely, high levels of integration among nurses and physicians in the team allows for better and more consistent management of patients’ demands, reducing the risk of conflict escalation. These results confirm the role played by social resources available in the work environment as a protective factor against the risk of third-party violence episodes in the healthcare setting [35].

Our findings imply that efforts to prevent workplace violence should also consider mitigating exposure to psychosocial risk factors at work because these factors fuel the experience of job stress in healthcare workers, compromising their capacity to effectively manage relationships with patients and their family members—a self-evident root cause of workplace violence. So far, different risk factors for workplace violence have been emphasized, including attitudes and behaviors of patients, work environmental factors such as shortages of staff and overcrowded or isolated areas of work, and characteristics of healthcare employees such as inexperience and lack of training [16,46]. Interventions on healthcare workers have frequently insisted on violence prevention training mainly based on de-escalation techniques. However, these interventions have overall been unsuccessful so far in reducing rates of occurrence of workplace violence [47]. The importance of acting on work-related stress and its correlates, such as burnout, which usually reaches very high levels among healthcare workers [30], has not yet received sufficient attention from researchers and institutions focused on workplace violence prevention.

The prevention of stress at work may be based on the management of modifiable work-related psychosocial risk factors, including those focused on in the present study [48]. First of all, since workload is an important factor capable of exacerbating the risk of violence, an increase in staffing should be carefully considered, as well as an equal distribution of caseloads among healthcare professionals. Additionally, workload can be managed by developing action plans, prioritizing demands, monitoring tasks on an ongoing basis, and reviewing processes to see if work can be improved. Such interventions can be implemented by both employees and supervisors, if properly trained, for example, in how to craft their job [49]. This may help make workload more manageable and reduce associated stressful experiences. Similarly, job control can be improved by providing more discretion to employees about how and when their work can be done [50,51]. This may mean offering predictability and flexibility in working hours, also considering extra work commitments, granting autonomy on how to approach work tasks, and providing the possibility of using a variety of job skills and making decisions that influence one’s work.

Supervisor support at work, a group-level resource that may have a positive impact on all team members at the same time, can be enhanced by developing specific management competences for preventing the experience of excessive stress in subordinate workers through specific training, such as workshops, peer and upward feedback, and self-reflection activities [52]. There have been developments in this area, and such interventions have been proposed as good practices for stress prevention and health promotion in the workplace [53]. Also, short daily meetings between staff and supervisor, focused on the clarification of activities and the discussion of patient-related problems, can strengthen perceived trust and support from supervisors [35]. Finally, team integration, another important contextual resource according to our results, can be strengthened by team-building activities, which have been shown to be effective in increasing teamwork and team performance in a variety of settings ad occupations [54]. Specifically, different healthcare professionals could be involved in common training sessions in order to develop a shared representation of patient issues, reduce divergent perspectives due to different professional roles, and improve communication and integration within the team [28]. To obtain enduring results from all these initiatives, healthcare organizations should adopt a ‘continuous improvement’ approach, whereby attention to work environmental conditions remains constantly high [48].

One of the main limitations of the present study is the cross-sectional nature of the data available, which prevents causal inferences regarding the examined relationships. For example, we cannot exclude that excessive patients’ demands may be a consequence of experiencing workplace violence rather than the reverse. However, the proposed model is in line with the so-called ‘work environment hypothesis’, which has been used to explain being the target of workplace mistreatment (i.e., a particular form of violence) from organizational insiders [55]. This model, which emphasizes the role of work environmental conditions in the escalation of workplace mistreatment, has been robustly validated with longitudinal studies [56]. Despite this, there is a need to conduct longitudinal investigations to probe the true causal ordering of the investigated phenomena. Additionally, we did not directly examine ‘objective’ workplace violence, such as official incidents of violence included in organizational records. Although objective report may underestimate actual episodes, self-reported violence is a common method to investigate the occurrence of this phenomenon [11]. Future studies should replicate our findings with a multi-method approach. However, we found a certain convergence between self-reported data on workplace violence and objective data available for a certain number of groups included in the present investigation. A final limitation is that some of the considered relationships might be more meaningful for certain groups of healthcare workers (e.g., physicians and nurses) than for others whose job activities do not involve direct care provision as the main part of the job (e.g., administrative staff and technicians).

## 5. Conclusions

The present study documents that psychosocial risk factors, acting at different levels and fueling the experience of job stress, play a critical role in workplace violence dynamics, particularly in workplace violence from patients and their family members, as reported by healthcare workers. Job stress and workplace violence prevention activities are usually seen as two separate issues in the area of health and safety at work, probably because job stress and workplace violence are not considered as manifestations—at least in part—of the same underlying causes. The present study suggests that job stress and workplace violence may share a poorly organized work environment, as indicated by higher workloads, lower levels of job control, little support from supervisors, low team integration, and probably other psychosocial risk factors that were not considered in the present analysis. Work-related stress and psychosocial risk factors can be assessed and effectively managed with specific primary, secondary, and tertiary preventive interventions aimed at reducing their impact on the health and well-being of workers [57]. By taking such activities seriously, healthcare organizations will not only protect the job performance of their workers but also ensure that the risk for the occurrence of workplace violence incidents from third parties, specifically patients and their family members, will be significantly mitigated. Given the substantial personal and organizational costs associated with workplace violence, the importance of reducing workers’ exposure to job stress should not be underestimated by healthcare organizations.

## Figures and Tables

**Figure 1 ijerph-21-00178-f001:**
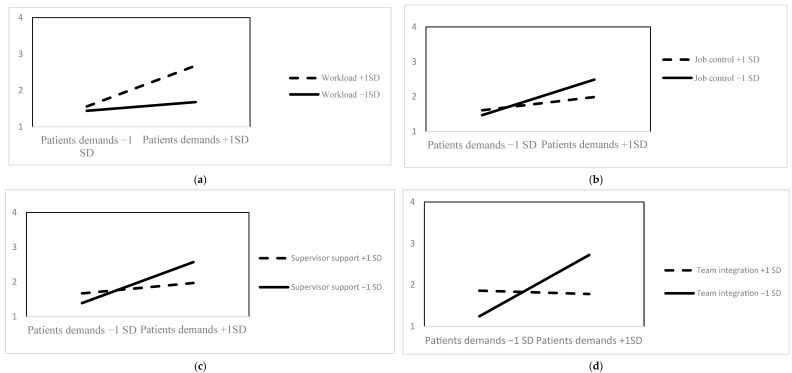
Graphical representations of the interaction effects between patients’ demands and workload (**a**), job control (**b**), supervisor support (**c**), and team integration (**d**). Dependent variable: workplace violence.

**Table 1 ijerph-21-00178-t001:** Pearson correlations between the main study variables.

	M (SD)	1	2	3	4	5	6	7
1-Workplace violence	2.02 (0.88)	-	0.40 ***	0.36 ***	−0.22 ***	−0.18 ***	−0.16 ***	-
2-Patients’demands	3.10 (0.77)	0.73 ***	-	0.35 ***	−0.14 ***	−0.09 *	−0.16 ***	-
3-Workload	2.94 (0.60)	0.60 ***	0.69 ***	-	−0.36 ***	−0.23 ***	−0.29 ***	-
4-Job control	3.17 (0.69)	−0.14	−0.44 ***	−0.46 ***	-	0.26 ***	0.37 ***	-
5-Supervisor support	3.29 (0.81)	−0.30 *	−0.35 ***	−0.24	0.21	-	0.55 ***	-
6-Equipe integration	3.64 (0.81)	−0.20	−0.42 ***	−0.34 *	0.47 ***	0.70 ***	-	-
7-Workplace violence (objective)	1.29 (1.28)	0.42 *	0.47 *	0.50	−0.01	−0.22	−0.27	-

Note. N = 681 employees. Variables 1–6 are based on self-reported data. Variable 7 is an index of ‘official’ workplace violence incidents for the available groups derived from data extracted from organizational records and computed as follows: (n. of violent incidents reported by workers in the group/n. of workers in the group) × 100. Within-group correlations—based on within-group centered variables are reported above the diagonal, while between-group correlations are reported below the diagonal. Correlations at the between-group level are based on 56 groups, except for the variable ‘Workplace violence (objective)”, for which N = 26. * *p* < 0.05; *** *p* < 0.001.

**Table 2 ijerph-21-00178-t002:** Reports the results of multilevel analysis.

	Null Model Estimate (*SE*)	Model 1 Estimate (*SE*)	Model 2 Estimate (*SE*)	Model 3 Estimate (*SE*)	Model 4 Estimate (*SE*)	Model 5 Estimate (*SE*)	Model 6 Estimate (*SE*)
Intercept	2.02 (0.08) ***	1.88 (0.15) ***	1.84 (0.15) ***	1.89 (0.15) ***	1.89 (0.15) ***	1.90 (0.15) ***	1.90 (0.15) ***
Gender (0 = male, 1 = female)		−0.03 (0.06)	-0.02 (0.06)	−0.02 (0.06)	−0.03 (0.06)	−0.03 (0.06)	−0.02 (0.06)
Age_1		−0.05 (0.11)	-0.04 (0.11)	−0.07 (0.11)	−0.05 (0.11)	−0.05 (0.11)	−0.05 (0.11)
Age_2		0.08 (0.10)	0.08 (0.10)	0.06 (0.10)	0.05 (0.10)	0.04 (0.10)	0.05 (0.10)
Role_1		−0.05 (0.10)	−0.05 (0.10)	−0.06 (0.10)	−0.03 (0.10)	−0.07 (0.10)	−0.05 (0.09)
Role_2		−0.14 (0.16)	−0.15 (0.16)	−0.15 (0.16)	−0.13 (0.15)	−0.15 (0.15)	−0.15 (0.15)
Night shift (0 = no, 1 = yes)		0.19 (0.07) **	0.19 (0.07) **	0.18 (0.07) *	0.19 (0.07) **	0.19 (0.07) **	0.19 (0.07) **
Emergency dept. (0 = no, 1 = yes)		0.91 (0.24) ***	0.93 (0.24) ***	0.94 (0.24) ***	0.82 (0.24) ***	0.83 (0.23) ***	0.85 (0.24) ***
Patients’ demands (PD)		0.35 (0.05) ***	0.34 (0.05) ***	0.35 (0.05) ***	0.36 (0.06) ***	0.37(0.05) ***	0.35 (0.05) ***
Workload (WL)		0.30 (0.05) ***	0.28 (0.05) ***	0.31 (0.05) ***	0.30 (0.05) ***	0.30 (0.05) ***	0.31 (0.05) ***
Job control (JC)		−0.10 (0.05) *	−0.11 (0.04) *	−0.09 (0.04)	−0.10 (0.04) *	−0.10 (0.04) *	−0.09 (0.04) *
Supervisor support (SS)		−0.02 (0.13)	−0.02 (0.13)	−0.02 (0.13)	−0.01 (0.13)	−0.08 (0.13)	−0.03 (0.13)
Equipe integration (EI)		−0.12 (0.18)	−0.10 (0.18)	−0.10 (0.18)	−0.08 (0.18)	−0.03(0.17)	−0.08 (0.18)
PD × WL			0.22 (0.07) **				
PD × JC				−0.16 (0.07) *			
PD × SS						−0.22 (0.07) **	
PD × EI							−0.39 (0.10) ***
Within group (L1) variance	0.56	0.43	0.42	0.42	0.40	0.40	0.40
Intercept (L2) variance	0.25	0.15	0.16	0.15	0.16	0.16	0.16
Slope (L2) variance					0.07	0.04	0.03
Intercept/slope (L2) covariance					0.04	0.04	0.03
−2 log lilkelihood (n. of parameters)	1583.51 (3)	1395.63 (15)	1385.83 (16)	1389.77 (16)	1386.00 (17)	1376.05 (18)	1372.79 (18)
∆ −2 log lh (∆ n. of parameters)		125.90 (5) ***	9.80 (1) **	5.86 (1) *	9.63 (2) **	9.95 (2) **	13.82 (2) ***

Note. The age and role covariates were modeledusing dummy variables coded as follows. Age_1: 1 = more than 50 years, 0 = up to 50 years; Age_2: 1 = 31–50 years, 0 = up to 30 years OR more than 50 years. Role_1: 1 = nurse, 0 = others. Role_2: 1 = doctor, 0 = others. * *p* < 0.05; ** *p* < 0.01; *** *p* < 0.001.

## Data Availability

Data are available upon request to the corresponding author (cristian.balducci3@unibo.it).

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
