# Peer review of "The Relationship between Patients’ Demands and Workplace Violence among Healthcare Workers: A Multilevel Look Focusing on the Moderating Role of Psychosocial Working Conditions"

_ijerph, 2024, doi:10.3390/ijerph21020178_

Round 1

Reviewer 1 Report

Comments and Suggestions for Authors

Line 44 - For all percentages write the word percent.  The symbol is used in tables and charts.  All single digit numbers should be written.

TABLES

All tables should be introduced separately in the text before the table indicating what can be found in the table.  A summary of what is found in the table should follow after each table. 

Comments on the Quality of English Language

English is fine.

Author Response

Please, see attached file. Thank you.

Reviewer 2 Report

Comments and Suggestions for Authors

I congratulate the authors on the topic and the context in which they carry out their research, which is undoubtedly very relevant.

However, there are some points that I think should be taken into account in order to improve the quality of the manuscript.

While it is true that the initial hypothesis of the research team is made explicit, it would be appropriate to include at least the general aim of the study. It is mentioned that it is part of a larger project, the aim of which is made explicit, but the aim of the presented study is not reflected. 

INTRODUCTION

I think it should be shortened. It is too long and its length is detrimental to that of the conclusions, which are too few.

RESULTS AND DISCUSSION

There is an error in the location of Table 2 and the referenced figures. They should be moved to the Results section. They are currently misplaced in the Discussion section.

In addition, the sex variable in Table 1 is on the wrong axis. The separation of data by sex is an essential requirement in the scientific literature. It must be placed on the horizontal axis in order to obtain the information for each of the variables analysed broken down by sex. This is an essential requirement for any study, but in the case of this one it is even more necessary so that the gender perspective can be included in the discussion, a measure that I suggest the authors implement.

CONCLUSIONS

The conclusions should be extended.

REFERENCES

I suggest that bibliographical references more than 7 years old be replaced or updated. The subject of the study is very well researched and there is a lot of more recent scientific literature.

Reviewer 3 Report

Comments and Suggestions for Authors

Dear Authors,  

The review deals with interesting and important research area analyzing occupational violence in healthcare sector. In my opinion, the manuscript is well structured, well-written, and provides clear information. However, some recommendations could improve its quality. I hope my comments and minor revisions would be helpful for improvement of the manuscript.

My comments:

1. Keywords:

“Equipe integration” is unclear (might be “equipped”?). It is excessive and could be excluded.

2.References:

-There are 70 references, what is too many for this kind of paper. Some of references are too old, so could be deleted.

-In addition, there are technical mistakes in the list of references.

For instance,: ref. 18 has duplicated 2013, 2013; some journal titles are presented as abbreviations, some – in full title; after title abbreviation should be a dot; some of authors’ name letter has a dot, some – haven’t, etc. Please, look carefully and correct.

Recommendation: Minor revision.
